# Acute Changes in Heart Rate Variability to Glucose and Fructose Supplementation in Healthy Individuals: A Double-Blind Randomized Crossover Placebo-Controlled Trial

**DOI:** 10.3390/biology11020338

**Published:** 2022-02-21

**Authors:** Max Lennart Eckstein, Antonia Brockfeld, Sandra Haupt, Janis Ramon Schierbauer, Rebecca Tanja Zimmer, Nadine Bianca Wachsmuth, Beate Elisabeth Maria Zunner, Paul Zimmermann, Maximilian Erlmann, Barbara Obermayer-Pietsch, Felix Aberer, Othmar Moser

**Affiliations:** 1Division of Exercise Physiology and Metabolism, Department of Sport Science, University of Bayreuth, 95440 Bayreuth, Germany; max.eckstein@uni-bayreuth.de (M.L.E.); antonia.brockfeld@uni-bayreuth.de (A.B.); sandra.haupt@uni-bayreuth.de (S.H.); janis.schierbauer@uni-bayreuth.de (J.R.S.); rebecca.zimmer@uni-bayreuth.de (R.T.Z.); nadine.wachsmuth@uni-bayreuth.de (N.B.W.); beate.zunner@uni-bayreuth.de (B.E.M.Z.); paul.zimmermann@uni-bayreuth.de (P.Z.); maximilian.erlmann@uni-bayreuth.de (M.E.); felix.aberer@uni-bayreuth.de (F.A.); 2Endocrinology Lab Platform, Division of Endocrinology and Diabetology, Department of Internal Medicine, Medical University of Graz, 8036 Graz, Austria; barbara.obermayer@medunigraz.at; 3Interdisciplinary Metabolic Medicine, Division of Endocrinology and Diabetology, Department of Internal Medicine, Medical University of Graz, 8036 Graz, Austria

**Keywords:** glucose, fructose, sucralose, cardio-autonomic response, heart-rate variability, hemodynamic changes

## Abstract

**Simple Summary:**

In this study, we investigated the cardio-autonomic stress responses to the ingestion of liquid glucose, fructose, a combination thereof and a placebo in healthy individuals at rest. The cardio-autonomic response was more pronounced in all groups with carbohydrates compared to placebo indicating an increased cardio-autonomic stress response resulting in a reduced heart-rate variability. When investigating different levels of blood glucose, the findings showed a significant decline in heart-rate variability with increasing blood glucose levels. This was also seen with severely low levels of blood glucose. The speed of how quick blood glucose increased and decreased also impacted the cardio-autonomic response which further deteriorated heart-rate variability. These findings indicate that healthy human’s autonomic system responds quickly to changes in their blood glucose.

**Abstract:**

Background: It is unknown how different types of carbohydrates alter the cardio-autonomic system in healthy individuals. Therefore, the aim of this study was to investigate how heart-rate variability changes to single dose ingestion of glucose, fructose, glucose and fructose, and an artificial sweetener (sucralose). Methods: In a double-blind randomized crossover placebo-controlled setting, 15 participants received all study-specific substances in liquid form. During each 2-h visit, venous blood glucose was measured in a 5-min interval while heart-rate variability was measured continuously via Holter-electrocardiograph. Results: Ingestion of different types of carbohydrates and sucralose showed significant differences for heart rate (*p* < 0.001), SDNN (*p* < 0.008), RMSSD (*p* < 0.001), pNN50 (*p* < 0.001) and blood pressure (*p* < 0.001). Different glucose levels significantly altered parameters of heart-rate variability and blood pressure (all *p* < 0.001), while the rate of change in blood glucose led to changes in heart rate variability, but not in heart rate (*p* = 0.25) or blood pressure (*p* = 0.99). Conclusions: Ingestion of different types of carbohydrates lead to reductions in heart-rate variability compared to a placebo. Blood glucose values above or below 70–90 mg/dL decreased heart rate variability while this was also seen for rapid glucose changes, yet not as pronounced. Healthy individuals should be conscious about carbohydrate intake while maintaining blood glucose levels between 70–90 mg/dL.

## 1. Introduction

Heart rate variability (HRV) is a widely used marker of the cardiac autonomic nervous system, well known for its versatile applicability in health and disease [1,2,3,4]. Previous research has shown that changes in HRV may indicate an increased risk of sudden cardiac death [5], atrial fibrillation [6], and cardiac autonomic neuropathy [7]. Particularly in individuals with impaired glucose metabolism, as it is the case in type 1 diabetes (T1DM) and type 2 diabetes mellitus (T2DM), the appearance of diabetic cardiac autonomic neuropathy is associated with chronically decreased HRV [8]. In case of diabetes mellitus, impaired cardiac autonomic modulation appears to be a progressive phenomenon that may be associated with dysglycemia with extended periods of hypoglycemia and hyperglycemia deteriorating overall glycemic control [9,10]. The relationship between blood glucose level and its impact on cardiac autonomic modulation, particularly the mechanisms and interactions thereof are yet to be understood [11,12,13]. Blood glucose ranges in individuals with diabetes are separated in hypoglycemia (<70 mg/dL, <3.9 mmol/L), euglycemia (70–180 mg/dL, 3.9–10.0 mmol/L) and hyperglycemia (>180 mg/dL, >10.0 mmol/L) [14]. The time spent in those glycemic ranges are decisive for good glycemic management and are therefore one of the indicators, besides the gold standard glycated hemoglobin HbA_1c_, for making therapy adaptations.

We have recently shown that in individuals with T1DM without cardiac autonomic neuropathy, rapid increases in blood glucose lead to significant short-term reductions in HRV [15]. This was somewhat surprising since previous research has merely focused on glycemic ranges. It can be suspected that not only glycemic ranges have an influence on the development of cardiac autonomic neuropathy but also the glycemic variability throughout the day. This variability induces an immediate reduction in HRV, which is reflective of an increased sympathetic stress response that over time may potentially develop into cardiac autonomic neuropathy [16].

In healthy humans, glucose metabolism rarely deserves any attention until by coincidence or due to specific symptoms T1DM or T2DM is diagnosed. Until then, any type or amount of carbohydrate is consumed without giving it a second thought regarding its influence on blood glucose levels. However, the type of consumed carbohydrates may have an impact on the physiology of the individual and may lead to an altered metabolic response [17]. In healthy humans, fasting blood glucose levels usually range between 72 to 108 mg/dL (4 to 6 mmol/L) [18]. Nonetheless, different amounts of carbohydrates may lead to blood glucose increments above 150 mg/dL or to blood glucose decrements below 70 mg/dL following an overexpression of pancreatic insulin [17]. From this perspective, it is of interest how the heart responds to rapidly altered blood glucose levels following the ingestion of different types of carbohydrates in healthy individuals. Since cardiac autonomic responses, detected via HRV have been subject of investigation for several decades under resting and exercise conditions [19,20,21], it is of interest how metabolic stress induces a cardiac autonomic response. To the best of our knowledge research has solely been conducted in individuals with T1DM and T2DM [12,15,22], investigating the effects of glycemia and HRV. Consequently, the aim of this study was to investigate how different types of fluid carbohydrates consumed via an individualized amount influence blood glucose levels and lead to a cardiac autonomic response measured via HRV in a double-blind randomized placebo-controlled fashion.

## 2. Materials and Methods

This study was performed as a single center, randomized, double-blind, placebo-controlled crossover trial investigating the effects of different types of carbohydrates and their effect on cardiac autonomic response measured via HRV. The local ethics committee of the University of Bayreuth (Bayreuth, Germany) approved the study protocol (O 1305/1.GB, 26 April 2021), which was registered at the German Clinical Trials Register (DRKS00024755). The study was conducted in conformity with the declaration of Helsinki and Good Clinical Practice. Before any trial related activities, potential participants were informed about the study protocol and participants gave their written informed consent.

### 2.1. Assessment of Eligibility

Inclusion and exclusion criteria were assessed by an investigator at the screening visit two weeks prior to the first trial visit.

### 2.2. Eligibility Criteria

The following eligibility criteria were defined: male or female, age between 18–65 years, body mass index (BMI) of 18.0–29.9 kg/m^2^ (both inclusive), normal overnight fasting blood glucose (BG) levels. Individuals were excluded if they were enrolled in a different study, received investigational medicinal products, had a blood pressure outside the range of 90–150 mmHg for systolic and 50–95 mmHg for diastolic after resting for five minutes in a supine position. Potential participants were excluded if they had a history of multiple and/or severe allergies to any trial related products. Furthermore, participants were not eligible for participation in the study due to language barriers precluding adequate understanding or cooperation. Participants with a history of any kind of metabolic disease were not eligible for the study. Furthermore, participants that were taking beta-blockers or had any cardiac defect that could influence the main outcome of the study were excluded.

Women of childbearing potential who were pregnant, breast-feeding or intended to become pregnant without using adequate contraceptive methods prior to or during the study visits were not included. Prior to inclusion the medical history of every participant was recorded by a health practitioner. For safety reasons every participant underwent a 12-lead ECG in supine position which was examined by a cardiologist to avoid any abnormalities that could potentially the outcome of this study. No participant included in this study was taking any medication that could have potentially influenced the outcome which was approved by two health practitioners (B.E.M.Z. and P.Z.)

### 2.3. Study Design

After inclusion in the study, participants were assigned to ascending numbers and allocated to the order in which the trial visits were conducted in a cross-over randomized fashion with the software Research Randomizer^®^ (1:1:1:1) [23]. Participants received 1 g/kg body mass (BM) of carbohydrates. These contained either glucose (Glu), fructose (Fru) or a mixture of 50% Glu and 50% Fru (GluFru). Sucralose (Suc) was given with a fixed amount of 0.2 g per dosage since its taste is about one hundred times sweeter than glucose and exceeding amounts of this artificial sweetener might exert some toxicity [24]. Suc was used as a placebo control to imitate the taste of the other study related products to avoid selection bias. Directly prior to each visit trial products were mixed in 300 mL of water in opaque shaker bottles by a researcher no other involved in the study. Shakers were then labelled with the participant’s study ID. Opaque shakers were given to participants directly prior to ingestion to avoid any bias. Between each visit, a minimum period of 48 h was maintained to ensure that participants fully recovered between visits and to ensure that multiple fasting periods had no impact on BM during the course of the study.

### 2.4. Trial Visits

Participants attended the research facility in the morning (7 a.m.) after their overnight fast. No caffeine-rich drinks or diet sodas were allowed within the 12 h fasting periods, solely water. Participants were not allowed to consume any alcoholic beverages within 24 h prior to the fasting periods. In addition, they had to refrain from any strenuous physical exercise within 24 h prior to each visit. At the beginning of each study visit, all participants had to fill in an international physical activity questionnaire in short form (IPAQ-SF) to monitor changes in exercise behavior and to potentially reschedule visits once physical exercise increased to maintain comparability between trial arms. After a 5-min resting phase, participants consumed the visit specific pre-mixed fluid. Participants were asked to consume the drink within one minute. Immediately prior to consuming the drink, a venous blood sample of 20 µL was taken from the antecubital vein to measure fasting BG. For the following 2 h of the measurement period, venous BG samples were measured in a 5-min interval. During the 2-h trial visits, participants remained in a supine position.

### 2.5. HRV Measurement

Over the course of each 2-h study visit, participants underwent electrocardiography with a Holter monitor (Faros 180; Bittium, Oulu, Finland) using one channel with a 1000-Hz sampling rate. The HRV measures evaluated in the time domain analysis included standard deviation of R-R intervals (SDNN), square root of the mean standard difference of successive R-R intervals (RMSSD) and percentage of pairs of R-R intervals with >50 ms difference (pNN50%). Power spectral analysis for the analysis of the frequency domain was conducted via Fast-Fourier Transformation in Cardiscope (Hasiba Medical GmbH, Graz, Austria). Low frequency/high frequency (LF/HF) and RMSSD are measures of the balance between parasympathetic and sympathetic activity. HRV values were assessed according to the guidelines published by the Task Force of the European Society of Cardiology and the North American Society of Pacing and Electrophysiology for the assessment of HRV [25]. Blood pressure was measured via an upper arm cuff (Medisana BU512, Neuss, Germany) in a 5-min interval throughout the entire measurement.

### 2.6. Blood Glucose Stratification

For the purpose of this study we stratified glucose levels according to recent position statements on BG monitoring and previously published data [14,15]. BG levels were merged with HRV data for analysis. Glucose levels were stratified for hypoglycemia level 2 (<54 mg/dL, <3.0 mmol/L), hypoglycemia level 1 (54–70 mg/dL, 3.0–3.9 mmol/L), euglycemia level 1 (70–90 mg/dL, 3.9–5.0 mmol/L), euglycemia level 2 (90–110 mg/dL, 5.0–6.1 mmol/L), euglycemia level 3 (110–130 mg/dL, 6.1–7.2 mmol/L), euglycemia level 4 (130–150 mg/dL, 7.2–8.3 mmol/L) and hyperglycemia level 1 (>150 mg/dL, >8.3 mmol/L). This approach is uncommon to what is normally applied in determining glycemic ranges in individuals with T1DM and T2DM. In healthy individuals no evidence besides aforementioned euglycemic range exists. Consequently, we have separated the glycemic ranges for more detailed analysis. Different rates of blood glucose change were analyzed as follows: decrease in BG >10 mg/dL within 5 min (D2), decrease BG of 5–10 mg/dL within 5 min (D1), stable glucose, defined as a change of <5 mg/dL within 5 min (N), increase in BG of 5–10 mg/dL within 5 min (I1) and increase in BG >10 mg/dL within 5 min (I2). Data was subsequently merged with the HRV data for statistical analysis.

### 2.7. Blood Sampling

Venous BG samples were collected with 20 µL capillaries from the antecubital vein and analyzed via a fully enzymatic-amperometric method (Biosen S-line, EKF Diagnostics, Barleben, Germany). The samples were collected in a 5-min interval during the course of each study visit. Once capillaries were collected, the samples were given into 1000 µL pre-filled Eppendorf tubes (EKF Diagnostics, Barleben, Germany) and were shaken for dilution. This ensured that the sample was stable and left to rest until being analyzed directly at the end of each study visit.

### 2.8. Randomization Procedure

In the morning prior to each visit, a researcher no other involved in the implementation of the trial, prepared the drinks with Glu, Fru, GluFru or Suc, dependent on the randomization, in opaque shaker bottles and labelled them with the participant’s ID. This procedure was conducted to avoid any kind of bias from the researchers or participants by seeing clearness/cloudiness of the drinks prior to ingestion.

### 2.9. Statistical Analysis

All data were assessed for normal distribution by means of the Kolmogorov–Smirnov normality test. Venous BG and variables of HRV were analyzed via repeated-measures one-way analysis of variance (RM-ANOVA), glucose stratified HRV data was calculated via mixed-effects model with Geisser–Greenhouse correction. Differences between groups, timepoints, and group × timepoint were calculated in this fashion. Tukey’s multiple comparisons test with individual variances were computed for each comparison via Prism Software version 8.0 (GraphPad, San Jose, CA, USA). Statistical significance was accepted at *p* ≤ 0.05.

## 3. Results

Fifteen individuals (five females, age of 25.4 ± 2.5 years, BMI of 23.7 ± 1.7 kg/m^2^ with a body mass 76.3 ± 12.3 kg) participated in all four study visits. No participant withdrew or had to be withdrawn from the study due to violation of specific measurement day criteria.

### 3.1. Type of Carbohydrates

Overall, the mean heart rates showed significant differences between Glu, Fru, GluFru and Suc (*p* < 0.001). Suc showed a significantly lower heart rate compared to Glu (59 ± 2 bpm vs. 63 ± 2 bpm, *p* < 0.001), Fructose (63 ± 2 bpm and GluFru 65 ± 2 bpm *p* < 0.001). Heart rate in GluFru was higher compared to Glu (*p* = 0.01) and Fru (*p* = 0.02). No significant difference was found between Glu and Fru (*p* = 0.95). SDNN showed significant differences between all groups (*p* < 0.001). Significant differences were found between GluFru and Fru (*p* = 0.01) and Suc (*p* = 0.001). No significant difference was found between Glu (59.9 ± 7.5) and Fru (62.5 ± 5.7, *p* = 0.21) and GluFru (58.3 ± 5.4, *p* = 0.69) and Suc (64.1 ± 5.8, *p* = 0.09). RMSSD showed significant differences between all groups (*p* < 0.001). GluFru (43.3 ± 5.2) showed lower RMSSD compared to Glu (50.3 ± 3.3, *p* < 0.001), Fru (50.0 ± 5.6, *p* < 0.001) and Suc (53.7 ± 5.5, *p* < 0.001). No significant difference between Glu and Fru (*p* = 0.99), Glu and Suc (*p* = 0.051) and Fru and Suc (*p* = 0.09) was found. pNN50 showed significant differences between all groups (*p* < 0.001). Significant differences were found between Glu (31.1 ± 2.9) and GluFru (23.9 ± 5.8, *p* < 0.001) and Suc (36.2 ± 4.7, *p* < 0.001). Fru (30.8 ± 5.1) showed significantly lower values compared to GluFru (*p* < 0.001) and Suc (*p* < 0.001). No significant difference was found between Glu and Fru (*p* = 0.99). Systolic blood pressure was significantly different between all groups (*p* < 0.001). Suc (131 ± 2) was significantly higher compared to Glu (129 ± 2, *p* = 0.01), Fru (129 ± 2, *p* = 0.02) and GluFru (128 ± 2, *p* < 0.001). No significant differences were found between the other groups. Glu (72 ± 2) showed significantly lower values in diastolic blood pressure compared to Fru (75 ± 2, *p* < 0.001), GluFru (74 ± 2, *p* = 0.001) and Suc (74 ± 2, *p* = 0.001). No statistical differences were found between Fru, GluFru and Suc. Details over time are shown in Figure 1.

### 3.2. Glucose Range Stratification

Parameters of HRV were stratified for pre-defined glycemic ranges. The results are shown in Table 1 and for easier understanding represented in Figure 2. Information given in Table 1 and Figure 2 show the predefined glycemic ranges ranging from clinically relevant hypoglycemia (<54 mg/dL) to blood glucose levels >150 mg/dL.

### 3.3. Glucose Rate of Change Stratification

Parameters of HRV were stratified for pre-defined glycemic rate of change. The results are shown in Table 2 and for easier understanding represented in Figure 3. Different rates of blood glucose change were analyzed as follows: decrease in BG >10 mg/dL within 5 min (D2), decrease BG of 5–10 mg/dL within 5 min (D1), stable glucose, defined as a change of <5 mg/dL within 5 min (N), increase in BG of 5–10 mg/dL within 5 min (I1) and increase in BG >10 mg/dL within 5 min (I2).

## 4. Discussion

To our knowledge, this was the first study that investigated the effects of different types of carbohydrates, glycemic ranges and glycemic rate of change on cardio-autonomic response via HRV in healthy individuals. Our findings indicate that the consumption of different types of carbohydrates induces a cardio-autonomic response compared to a placebo. In addition, we found significant findings once results were diverted into glycemic ranges and glycemic rate of change which has not been shown before in healthy individuals.

Glu, Fru and GluFru showed significantly higher heartrates over the course of each study visit compared to Suc independent of the amount of 300 mL. Post-prandial increments in heart-rate, cardiac output and stroke volume have been shown previously in relation to the meal content of fat, protein and carbohydrates [26]. The underlying mechanisms are unclear, but may potentially be attributed to the body’s response to ingested glucose [27]. The initiated cascade through the action of insulin and its vasodilatory properties may also induce sympathetic activity. This could also be confirmed by parameters of cardio-autonomic response in our study since, SDNN, RMSSD and pNN50 were lower after consumption of Glu, Fru or GluFru in comparison to the placebo. In addition, changes in systolic and diastolic blood pressure were also found, however, as demonstrated in Figure 1, these changes are significantly different between groups. However, the clinical importance is negligible and coincides what was shown by Heseltine et al. and by Sauder et al. that highlighted that independent of meal content hemodynamic changes are only partly affected [28,29].

Independent of the type of carbohydrates consumed, we found significant differences once BG levels were stratified to pre-defined ranges. In T1DM and T2DM, low glucose levels begin with values below 70 mg/dL and turns into clinical relevant hypoglycemia once below 54 mg/dL according to recent guidelines [14]. In individuals with T1DM or T2DM these glycemic levels induce for instance discomfort, increased heart rate and impaired cognitive function. However, the euglycemic range is determined between 70 to 180 mg/dL. In healthy individuals the fasted recommended range is between 72 and 108 and mg/dL, which can be still considered as very wide. Our findings support the idea of narrowing glycemic ranges more precisely since significantly decreased levels in SDNN, RMSSD and pNN50 were found below and above 70–90 mg/dL. This indicates a more metabolically induced response leading to a significant increase in cardio-autonomic stress. This is entirely contrary to what was seen previously in individuals with T1DM recently shown by our study group [15]. Neither heart rate, SDNN, RMSSD nor pNN50% were significantly altered following an oral glucose tolerance test in that study [15]. It can be suspected that individuals with T1DM spend more time in either hypo- or hyperglycemia compared to healthy individuals due to the complex insulin management. Once the body has accustomed to higher or lower glucose levels, a sympathetic stress response is blunted due to previous ‘habituation’ [30]. In healthy individuals the sympathetic response to dysglycemia is a lot more pronounced, hence HRV is reduced. The cardio-autonomic stress response is also reflected by the hemodynamic changes represented by systolic and diastolic blood pressure. Hemodynamic levels above BG of 150 mg/dL were all significantly higher compared to other glycemic ranges. Once reaching these elevated BG levels as a healthy individual would indicate the highest insulin level that initiates a sympathetic cascade influencing HRV and blood pressure (Figure 2.). The strong correlation between insulin sensitivity and baroreflex gain shown by Young et al. in healthy men supports our findings [31]. Even though statistically significant the clinical relevance from some of our findings cannot be turned over to clinical significance as the change in heart rate over all glycemic ranges is clinically negligible.

A further result from our study was the cardio-autonomic response to the rate of BG change. Even small changes in BG that are not within ±5 mg/dL within 5 min lead to a cardio-autonomic response. As can be seen in Table 2 and Figure 3, SDNN, RMSSD and pNN50% decrease once BG values decrease or increase more rapidly than the physiological small variation in BG (N). It could be argued that these responses are closely associated to the level of glycemia, since rapidly decreasing BG values from hyperglycemia may be considered as comforting for the cardio-autonomic system leading to a decreased sympathetic tone and vice versa. However, blood pressure did not show any significant differences since the hemodynamic system may not respond as rapid as the cardio-autonomic system to small changes [32]. Again, this is left to be elucidated in humans of how hemodynamic responses change once BG increases or decreases rapidly for an extended period of time.

Our study is not without limitations, since it was conducted as a proof-of concept pilot study with an insufficient power-analysis due to the lack of published data in this field. A small increase in the sample size may further emphasize our findings and lead to clearer results when differentiating between the different types of carbohydrates and glycemic ranges. Even though our cohort is homogenous in BMI, age and fitness status a matched-cohort to differentiate between men and women would deliver more information and offer a deeper insight into the cardio-autonomic stress response to acute changes in glycemia. We did not measure HbA_1c_ in our non-funded study but all participants were examined by an experienced physician and fasting and feasted BG values were measured repeatedly prior to inclusion in the study to ensure that no individual had any form of (pre)-diabetes. Physically active individuals must not necessarily have a low HbA_1c_, which, dependent on the amount of daily carbohydrate intake and muscle mass, may be higher than the average population [33]. Once the body has accustomed to regularly rapid increases and higher levels of BG the body may have habituated to these circumstances and the cardio-autonomic response could be biased. Splitting healthy individuals into different quartiles of normal HbA_1c_ levels and investigating their responses similarly to what was performed in this study would deliver valuable results.

The findings from our study highlight that small changes in BG lead to no change in cardio-autonomic response. However, once the consumed amount of carbohydrates was increased the body responds with an increased sympathetic stress reaction. This also applies to low BG values, since healthy individuals show highest values of HRV within 70–90 mg/dL. Therefore, the BG recommendations that are contemplated as ‘healthy’ should be reevaluated with larger sample sizes.

## 5. Conclusions

Glu, Fru and GluFru lead to reductions in HRV compared to a placebo. Cardio-autonomic responses are decreased with BG levels above or below 70–90 mg/dL. The sympathetic activation also induces an increase in blood pressure once BG exceeds 150 mg/dL. Rapid changes in BG also lead to a deterioration of HRV, yet not as pronounced. Healthy individuals should be aware of the type and the amount of carbohydrates they consume since the body immediately responds with an increased sympathetic reaction.

## Figures and Tables

**Figure 1 biology-11-00338-f001:**
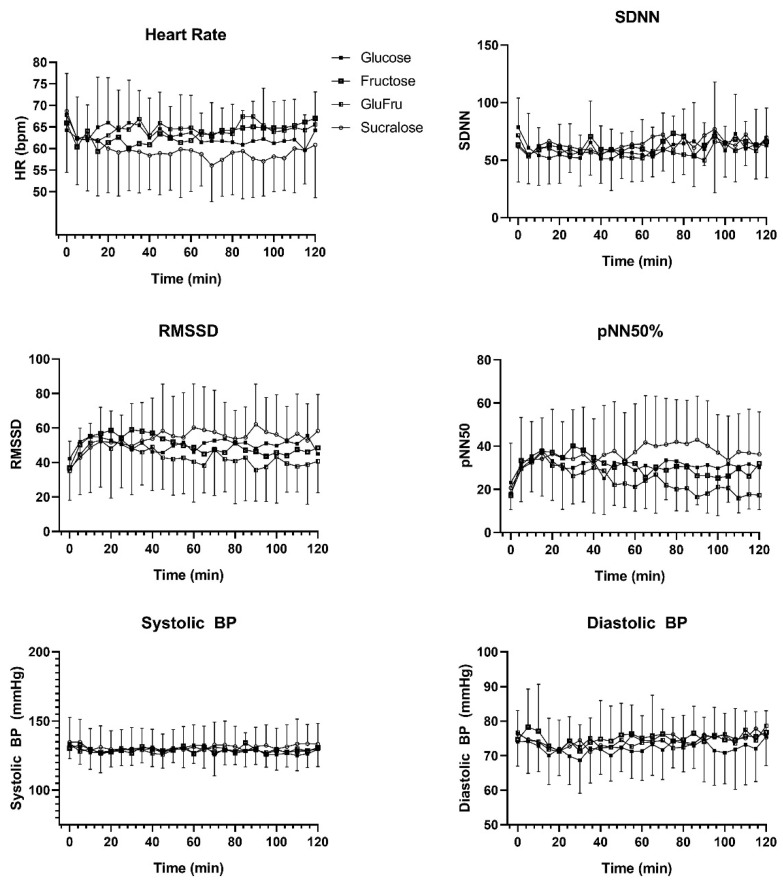
Change over time following the ingestion of glucose, fructose, glucose and fructose, and sucralose. Black squares indicate glucose, open squares indicate fructose, half-open squares indicate glucose and fructose, open circles indicate sucralose.

**Figure 2 biology-11-00338-f002:**
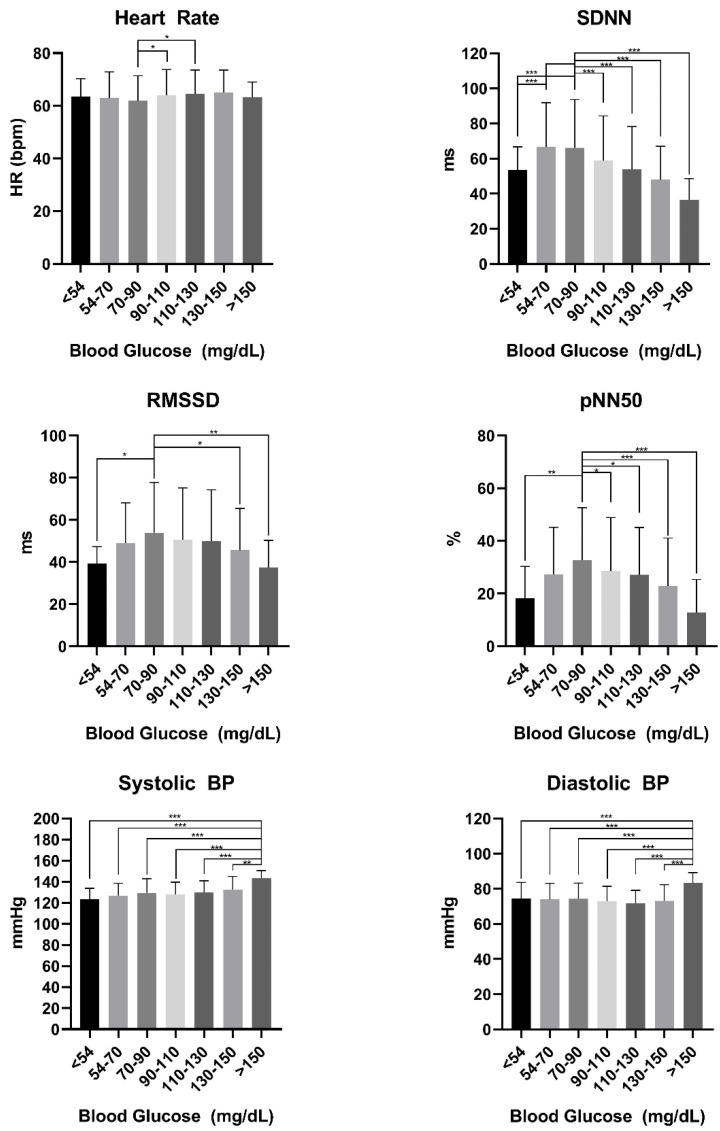
Indicates the glycemic range. BP: Blood pressure. ms: Milliseconds * indicates *p* < 0.05. ** indicates *p* < 0.01, *** indicates *p* < 0.001.

**Figure 3 biology-11-00338-f003:**
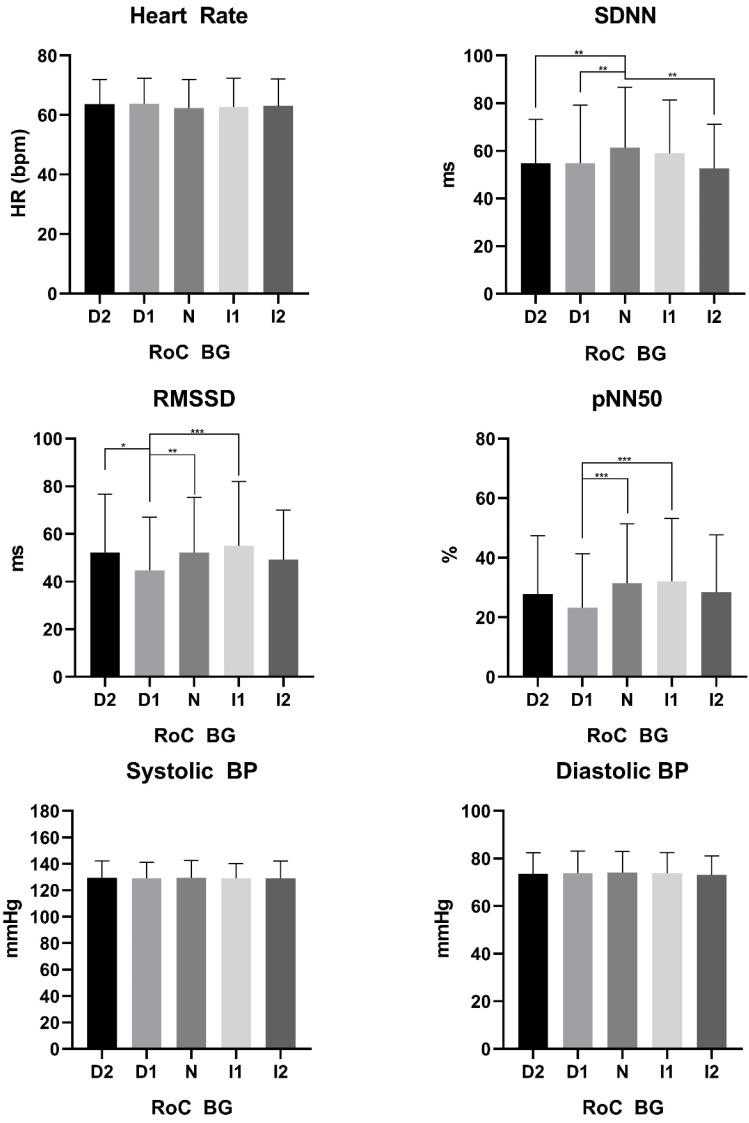
Indicates the rate of change (RoC) in BG. D2: Decrease in BG >10 mg/dL within 5 min. D1: Decrease BG of 5–10 mg/dL within 5 min. N: Stable glucose, defined as a change of <5 mg/dL within 5 min. I1: Increase in BG of 5–10 mg/dL within 5 min (I1) I2: Increase in BG >10 mg/dL within 5 min. ms: Milliseconds BP: Blood pressure. * indicates *p* < 0.05. ** indicates *p* < 0.01, *** indicates *p* < 0.001.

**Table 1 biology-11-00338-t001:** Indicates pre-defined glycemic ranges.

	<54	54–70	70–90	90–110	110–130	130–150	>150	*p*-Value
Heart Rate	63 ± 7	63 ± 10	61 ± 10	64 ± 10	65 ± 9	65 ± 8	66 ± 5	0.001
SDNN	53.4 ± 13.3	66.7± 25.2	66.1 ± 27.6	58.8 ± 25.6	53.9 ± 24.2	47.9 ± 19.2	36.5 ± 12.1	<0.001
RMSSD	39.3 ± 7.8	48.9 ± 19.2	53.9 ± 23.8	50.5 ± 24.7	50.0 ± 24.3	45.6 ± 19.9	37.4 ± 12.9	<0.001
pNN50	18.2 ± 12.0	27.3 ± 17.9	32.6 ± 19.9	28.52 ± 20.4	27.2 ± 17.9	22.9 ± 18.1	12.8 ± 12.5	<0.001
Systolic BP	124 ± 10	127 ± 12	130 ± 13	128 ± 12	130 ± 11	133 ± 12	143 ± 7	<0.001
Diastolic BP	74 ± 10	74 ± 9	73 ± 8	72 ± 7	72 ± 7	73 ± 9	83 ± 5	<0.001

Blood glucose values are displayed as mg/dL. Heart rate in beats per minute. SDNN: Standard deviation of NN intervals in milliseconds (ms). RMSSD: Root mean squared of adjacent NN intervals in ms. pNN50: The number of pairs of successive NN (R-R) intervals that differ by more than 50 ms. in %. Systolic BP in mmHg. Diastolic BP in mmHg.

**Table 2 biology-11-00338-t002:** Indicates the rate of glucose change.

	D2	D1	N	I1	I2	*p*-Value
Heart Rate	64 ± 8	64 ± 9	62 ± 10	63 ± 10	63 ± 9	0.25
SDNN	54.7 ± 18.5	54.8 ± 24.4	61.4 ± 25.3	58.9 ± 22.4	52.7 ± 18.5	<0.001
RMSSD	52.4 ± 24.3	44.7 ± 22.3	52.2 ± 23.0	55.1 ± 26.8	49.3 ± 20.8	<0.001
pNN50	27.9 ± 19.5	23.2 ± 18.2	31.5 ± 19.9	32.2 ± 21.0	28.4 ± 19.3	<0.001
Systolic BP	129 ± 13	129 ± 12	129 ± 13	129 ± 11	129 ± 12	0.99
Diastolic BP	74 ± 9	74 ± 9	74 ± 9	73 ± 9	73 ± 7	0.80

D2: Decrease in BG >10 mg/dL within 5 min. D1: Decrease BG of 5–10 mg/dL within 5 min. N: Stable glucose, defined as a change of <5 mg/dL within 5 min. I1: Increase in BG of 5–10 mg/dL within 5 min (I1) I2: Increase in BG >10 mg/dL within 5 min.

## Data Availability

Data will be made available upon reasonable request by the corresponding author.

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
