# Peer review of "Acute Changes in Heart Rate Variability to Glucose and Fructose Supplementation in Healthy Individuals: A Double-Blind Randomized Crossover Placebo-Controlled Trial"

_biology, 2022, doi:10.3390/biology11020338_

Round 1

Reviewer 1 Report

  1. When p<0.001, use <0.001, for example <0.0001 and <0.008 should be changed throughout
  2. Units of measurement should be provided for each variable table 1. 
  3. Abbreviations should be provided under table 1
  4. Unit of blood pressure and variable should also be provided in all figures
  5. The details on Randomization and study designs need to be additionally described.  

  6. How are the blinding process, need to explain concealment

  7. This study has not been registered in ClinicalTrials.gov?

  8. What method is being used to ensure allocation concealment?

  9. Has this study been approved by Institutional Review Board? Please provide details. 

Reviewer 2 Report

The manuscript entitled „Acute changes in heart rate variability to glucose and fructose supplementation in healthy individuals: a double-blind randomized crossover placebo-controlled trial” written by Max L. Eckstein et al. presents interesting results that reveal changes in heart rate variability after glucose and fructose ingestion in healthy individuals. The results obtained indicate how carbohydrates supplementation influences heart function. The reviewed work is properly designed and technically sound. The text is well written and the English language is appropriate and understandable. In the following I provide my suggestions for improving the manuscript.

General concept comments:

  1. Although detailed information about the study population was provided, there is little information about the cardiac examinations of the participants. In lines 113-115 it was only shortly mentioned that patients with cardiac defects were excluded from the study, but there is no information how the examination of the subject was carried out and what cardiac defects were evaluated. Patients taking beta-blockers were also excluded, but there is no information about taking other medications that could influence the cardio-autonomic system. Furthermore, past cardiac events (ex. myocardial infarction) and diseases could also affect the results of the study, therefore analysis of the medical history of the participants seems to be important during inclusion to the study. Please provide more information about these points as they are crucial when healthy individuals with normal heart function were assumed to be included in the study.
  2. In the manuscript text, sections 3.2. and 3.3. include almost only tables and figures. Authors should at least briefly comment on presented tables and figures to pay attention of readers on the most relevant results.
  3. In the text of the Result section, significant changes are described. However, the authors focused only on statistical significance, but the clinical significance of the results should also be evaluated, especially since some statistically significant results seem to have low clinical significance (ex. heart rate in Table 1).

Specific comments:

  1. Lines 59-60 – it was not indicated what type of glucose measurement is referring to the mentioned values of blood glucose levels – fasting or random tests? Please add such an information.
  2. Line 106 – here is the first time when ‘BG’ abbreviation was used and must be explained.
  3. Line 246 – it seems that “…table 1…” should be changed to “…table 2…” and “…figure 2…” should be changed to “…figure 3…”.
  4. In table 2 and figure 3, the identificators of rates of glucose change (D2, D1, N, I1 and I2) should be explained.
  5. Line 280 – “…and…” should be removed.

I believe that my suggestions will be helpful to the authors to increase the quality of the reviewed work.
